# Health-Related Gender Knowledge: Scale Development and Validation in Spanish Nursing Students

**DOI:** 10.3390/nursrep15060187

**Published:** 2025-05-27

**Authors:** Sara Prego-Jimenez, Jone Aliri, Olatz Goñi-Balentziaga, Eva Pereda-Pereda, Ainitze Labaka

**Affiliations:** 1Department of Clinical and Health Psychology and Research Methodology, Faculty of Psychology, University of the Basque Country (UPV/EHU), 20018 San Sebastian, Spain; 2Donostia University Hospital, Doctor Begiristain Avenue, 109, 20014 San Sebastian, Spain; 3Vitoria-Gasteiz Nursing School, Osakidetza-Basque National Health Service, Jose Atxotegi s/n, 01009 Vitoria-Gasteiz, Spain; eva.pereda@ehu.eus; 4Nursing II Department, Faculty of Nursing and Medicine, University of the Basque Country (EHU/UPV), 20014 San Sebastian, Spain

**Keywords:** gender bias, gender knowledge, gender sensitivity, nursing, health students, instrument development, psychometric validation, nursing education

## Abstract

**Background/Objectives:** Gender bias in healthcare remains a persistent challenge, partly due to gaps in gender-related knowledge among professionals. While existing instruments assess gender sensitivity and gender-role ideology, there is a lack of generalizable tools specifically designed to evaluate gender-related health knowledge. This study aimed to develop and validate the Gender Knowledge Scale in a sample of 591 nursing students from the University of the Basque Country (Spain). **Methods:** The 10-item multiple-choice instrument was created using the Nominal Group Technique with a multidisciplinary panel of experts. **Results:** Psychometric analyses supported a unidimensional structure with acceptable fit indices (CFI = 0.928, RMSEA = 0.025), and items displayed a diverse range of difficulty levels. Knowledge scores were moderately correlated with gender sensitivity but not with gender-role ideology, suggesting that knowledge may influence attitudes but is insufficient to challenge entrenched stereotypes. Knowledge differences emerged across academic years, though not by gender. Misconceptions, particularly regarding menstruation, cardiovascular risk, and pain, were prevalent. **Conclusions:** The Gender Knowledge Scale is a practical and psychometrically sound tool for assessing gender-related health knowledge among nursing students. While further validation is needed in other populations, the scale may support educational interventions aimed at reducing gender bias in clinical care.

## 1. Introduction

Research on gender bias in health care is a topic of broad interest and increasing attention in health and social science over the past decades. Differences in care provision [1,2], higher morbidity and mortality rates among female patients for various conditions [3,4], and the continued undermining and discomfort experienced by female patients have been widely explored and documented [5].

In this context, gender awareness and attitudes among health professionals have become a key area of research [6,7]. Gender awareness and knowledge are essential competencies in training health professionals and students [6,7,8]. In response, the World Health Organization [9] published a meeting report to promote the inclusion of the gender perspective in health-related curricula. Gender awareness has been defined as the “ability to view society from the perspective of gender roles and understand how this has affected women’s needs in comparison to the needs of men”. Within healthcare, gender awareness is often conceptualized through three core dimensions: (1) Gender role ideology (GRI), which refers to the extent to which a healthcare professionals make judgments based on gender-related stereotypes; (2) Gender sensitivity (GS), the extent to which they acknowledge and respond to gender-based health risks and needs [10]; (3) Knowledge, the extent to which professionals possess accurate information about gender and health [11].

Salgado et al. [11] developed the Gender Awareness Inventory-VA (GAI-VA), which included these three dimensions, specifically adapted to the Veterans Health Administration context. Subsequently, Verdonk et al. [10] developed the Nijmegen Gender Awareness in Medicine Scale (N-GAMS), which retained the GRI and GS dimensions but excluded the knowledge subscale.

Since sex and gender are recognized determinants of health, ignorance of their differential effects—and a lack of gender sensitivity—can perpetuate discriminatory attitudes and behaviors in healthcare [6,7].

From a cognitive perspective, schema theory suggests that individuals develop mental structures (schemas) to organize and interpret information [12,13]. In healthcare, these schemas influence how professionals perceive and respond to patient characteristics, including gender. Research in medical education has shown that such cognitive structures shape clinical reasoning and can perpetuate biases if not grounded in accurate knowledge [14]. In parallel, implicit bias theory highlights the role of unconscious attitudes in shaping healthcare behavior, especially under conditions of time pressure or uncertainty [15]. Strengthening accurate, reflective, and conscious gender-related schemas through education may thus contribute to more equitable care and reduce gender bias in clinical judgment [7].

The literature supports the link between healthcare professionals’ knowledge and their attitudes toward patients. For instance, Joukar et al. [16] found that professionals with higher levels of knowledge showed lower levels of discriminatory attitudes toward patients with hepatitis C. Similarly, Dayapoğlu and Tan [17], in a study of nurses’ knowledge and attitudes toward patients with epilepsy in Turkey, reported that greater knowledge was positively associated with more favorable attitudes and less discrimination.

According to this literature, gender knowledge plays a pivotal role in addressing gender bias and should, therefore, be central in research on gender awareness and clinical attitudes. However, the knowledge subscale developed by Salgado et al. [11] is context-specific and limited to the Veterans Health Administration. To date, no validated tools exist to assess gender knowledge in broader and more generalizable healthcare contexts.

This study aimed to develop and validate a new instrument to assess gender-related health knowledge. The tool offers a more comprehensive assessment of gender awareness by addressing the knowledge dimension omitted in widely used scales such as N-GAMS. It also allows for the evaluation of gender knowledge among nursing students and healthcare professionals and its potential relationship with discriminatory attitudes and behaviors.

The specific aim of this study was to develop the Gender Knowledge Scale and analyze its psychometric properties in a sample of Spanish nursing students.

## 2. Materials and Methods

### 2.1. Participants

A convenience sample of 591 Spanish nursing students participated in this study (259 first-year, 215 second-year, 52 third-year, and 65 fourth-year students). All were enrolled in the Faculty of Medicine and Nursing at the University of the Basque Country (UPV/EHU), Spain. Participants ranged in age from 18 to 49 years (M = 21.16; SD = 5.18), and the majority (85.4%) identified as female.

Given the number of items on the scale (*n* = 10), a minimum sample size of 100 participants was considered sufficient for initial scale validation [18]. However, to enhance the sample’s representativeness, the sample was expanded to approximately 600 participants.

### 2.2. Instruments

Several instruments were used, including the newly developed Gender Knowledge Scale, which is the focus of this study.

*Gender Knowledge Scale*. Gender-related knowledge was assessed using a self-administered questionnaire specifically designed for this study (see Section 2.3). The instrument consists of 10 multiple-choice questions, each with four answer options: one correct, one “I don’t know”, and two incorrect options reflecting common gender-related misconceptions. For example, the question “Choose the correct answer concerning pain” includes incorrect responses such as: “Women require less analgesia as they are physiologically prepared to tolerate pain”, and “Sex and gender have no influence on pain”.

Responses of “I don’t know” were excluded from scoring to calculate the total score. Correct answers received one point, while incorrect answers were penalized with −0.33. This scoring method is an adaptation of the correction-for-guessing method, aiming to discourage random responses while accounting for partial knowledge. The maximum possible score was 10, with higher scores reflecting greater gender-related health knowledge.

*Nijmegen Gender Awareness in Medicine Scale*. The brief Spanish adaptation of the Nijmegen Gender Awareness in Medicine Scale (S-NGAMS) [8] is a self-report questionnaire that assesses gender awareness in healthcare. It includes 25 items rated on a 5-point Likert scale, from “totally disagree” to “totally agree”. Fourteen items assess gender sensitivity (GS) (e.g., “Physicians’ knowledge of gender differences in illness and health increases the quality of care”), and eleven items assess gender-role ideology toward patients (GRI-patient) (e.g., “Female patients compared to male patients have unreasonable expectations of physicians”). Higher GS scores indicate greater sensitivity, whereas higher GRI-patient scores reflect stronger gender stereotypes. The Spanish version has demonstrated good psychometric properties, with a two-factor structure aligned with the original instrument and high internal consistency (α = 0.80 for GS, α = 0.89 for GRI-patient) [8]. In the present study, similar internal consistency values were obtained (α = 0.79 for GS, α = 0.88 for GRI-patient).

Additionally, a brief demographic questionnaire was included to collect data on age, gender, and academic year.

### 2.3. Procedure

The Gender Knowledge Scale was developed using the Nominal Group Technique (NGT), a structured method commonly used in health professions education research [19]. Experts were selected via purposive sampling based on their professional experience and formal training in gender and health. All had clinical practice and/or health education backgrounds and had completed training in gender perspectives. Experts were invited personally by the research team.

Two online sessions were held two weeks apart. The sessions were moderated by a member of the research team experienced in qualitative research and group facilitation. All nine invited experts attended both meetings. The panel included four nurses, three psychologists, a primary care physician, and a patient affiliated with an association for the promotion of women’s health.

In the first session, panelists responded in silence to the question: “Which gender-related topics are essential to assess gender-sensitive health knowledge in health professionals?” Their responses were collected and listed for discussion. After clarification and group discussion, each panelist privately rated the importance of each topic using a 1–9 scale (1–3 = not essential, 4–5 = neutral, and 7–9 = essential). Topics with a combined score > 63 were retained, and included: (a) basic gender perspective concepts in medicine, (b) epidemiology of sex differences, (c) gender differences in psychotropics prescription, (d) clinical assessment bias, and (e) under/overestimation of diseases by gender in the collective imagination. Panelists were then asked to draft 10 multiple-choice questions for the next session.

In the second session, a consolidated list of 23 draft questions was reviewed. Items considered ambiguous, redundant, overly specific, or too easy were eliminated or revised. Panelists then ranked the items again, and the 10 highest-rated questions were selected for the final version. Each question included one correct option, one “I don’t know”, and two incorrect options. The inclusion of the “I don’t know” choice was intended to minimize random guessing and allow for the expression of uncertainty.

Participants were recruited via institutional email. The questionnaire was distributed during the first semester of the academic year. Participation was voluntary and anonymous. Prior to completion, students received an information sheet explaining the study and were required to provide informed consent. Ethical approval was obtained from the university’s ethics committee.

Data collection was conducted online. The completion of the questionnaire took approximately 15 min.

### 2.4. Data Analysis

We calculated item-level difficulty indices and the percentages of correct, incorrect, and “I don’t know” responses. A confirmatory factor analysis (CFA) using the WLSMV estimator was conducted with the *lavaan* package (v 0.6-17) in R to assess the scale’s unidimensionality. The 10 dichotomous items were treated as categorical variables. To evaluate item discrimination, we compared correct and incorrect responses between participants with low (≤30th percentile) and high (≥70th percentile) total scores on the Gender Knowledge Scale. Non-parametric Kruskal–Wallis and Mann–Whitney U tests were used to explore differences in knowledge scores by academic year and gender. Finally, Spearman correlations were calculated between gender knowledge scores and the GS and GRI-patient subscales to gather validity evidence based on relations with other variables. All analyses were conducted using SPSS 28 (version R 4.3.3).

## 3. Results

### 3.1. Item Analysis

As shown in Table 1, the items varied considerably in difficulty. Specifically, five items were difficult or very difficult, with correct response rates below 38%. Three items were of medium difficulty, with correct response rates of around 60%. Finally, two items were easy or very easy, with correct response rates above 74%. The percentage of incorrect answers was very high in three of the difficult questions and relatively low in the easier ones. It is noteworthy that while the “I don’t know” option was frequently chosen for some difficult items, this was not consistent across all of them. The exact frequencies of each incorrect option are provided in Appendix A.

The one-factor model showed an adequate fit to the data: CFI = 0.928, TLI = 0.908, RMSEA = 0.025 (90% CI = 0.000–0.041), and SRMR = 0.075. These indices support the unidimensional structure of the scale. As shown in Table 1, most items presented moderate standardized factor loadings (λ = 0.27–0.55), indicating their contribution to the latent construct. Items 9 and 10 showed low loadings (0.12 and −0.04, respectively), suggesting that their performance may need to be revisited in future applications of the scale.

### 3.2. Item Discrimination and Relationship with Other Variables

The comparison between high and low scorers on the Gender Knowledge Scale revealed that the percentages of correct responses were consistently higher and incorrect responses lower in the high-scoring group across all items. As shown in Table 2, these differences were statistically significant (*p* < 0.001) and ranged from moderate to relatively strong in magnitude.

Regarding academic year, there were statistically significant differences in total knowledge scores (Kruskal–Wallis *H*(3) = 28.06, *p* < 0.001, *η*^2^ = 0.043), although the effect sizes were small. Pairwise comparisons using Dunn’s test with Bonferroni correction indicated that first-year students scored significantly lower than those in second (*p* = 0.001, *r* = 0.21) and fourth year (*p* = 0.026, *r* = 0.12).

No statistically significant differences were found between male and female participants (Mann–Whitney U = 19,579, *p* = 0.258, *r* = 0.05).

Finally, the correlation between gender knowledge and gender sensitivity was statistically significant and of moderate magnitude (rho = 0.30; *p* < 0.001), while the correlation between knowledge and gender-role ideology toward patients was small and not statistically significant (rho = 0.01; *p* = 0.802).

## 4. Discussion

The results of the present study indicated that the Gender Knowledge Scale is a reliable instrument for assessing health-related gender knowledge in nursing students. The varying difficulty levels of the items demonstrate the scale’s ability to differentiate between superficial and in-depth knowledge. Moreover, total scores on the scale were significantly correlated with the gender sensitivity subscale, supporting the idea that education aimed at increasing conceptual knowledge may influence health professionals’ attitudes and help prevent discriminatory behaviors [16].

However, no significant correlation was found between gender knowledge and gender-role ideology toward patients. This may suggest that conceptual knowledge alone is insufficient to challenge deeply rooted gender stereotypes. While increased knowledge can contribute to more sensitive attitudes, stereotypical beliefs may be more persistent and shaped by broader socialization processes. Reducing such thinking may, therefore, require not only knowledge acquisition but also targeted interventions that promote critical reflection and self-awareness. These findings are consistent with theoretical frameworks such as schema theory, which posits that clinical decision-making is influenced by internalized mental representations shaped by both knowledge and experience. Additionally, implicit bias theory highlights how unconscious stereotypes may persist despite formal instruction, emphasizing the need for pedagogical strategies that combine conceptual content with critical reflection.

In addition to the evidence based on external correlations, the confirmatory factor analysis supported the structural validity of the scale. The one-factor model showed acceptable fit indices, reinforcing the interpretation of the scale as a unidimensional measure of gender-related health knowledge. As shown in Table 1, most items presented moderate standardized loadings, while items 1, 9, and 10 showed lower values. This may be partially explained by their extreme difficulty or ease, which limits their statistical contribution. Nevertheless, we choose to retain them, as they help to differentiate individuals with particularly low or high levels of knowledge.

We also found significant differences in knowledge level between first- and second-year students. The higher scores among second-year participants likely reflect the integration of gender equity content in the first-year course, *Anthropology*, *Ethics, and Legislation*. However, no further improvements were observed in more advanced courses. This stagnation may indicate a lack of integration of the gender-related content as a transversal and continuous learning objective throughout the degree. Some authors have argued that, despite political mandates to include the gender perspective in health science curricula, this objective is still far from being reflected in newly graduated health professionals’ clinical competencies [3]. The knowledge gaps observed, particularly in items related to pain perception, cardiovascular risk, or premenstrual disorders, may stem from the absence of these topics in the formal curriculum or from how they are addressed in a non-specific or insufficient manner. Although second-year students performed better than first-year students, the absence of progression in later years may suggest that this content is not systematically revisited or reinforced. We believe that to address this gap, gender-related content should be integrated longitudinally throughout the nursing curriculum, not only as a foundational competence in early courses but also through its application in clinical subjects, case-based learning, and reflective practice.

Regarding gender, no significant differences in knowledge were found between women and men students. Similarly, Dielissen et al. [19] found no differences between female and male general practitioner trainees in a true-or-false knowledge test about gender-specific medical conditions.

The analysis of correct, incorrect, and “don’t know” responses was helpful in determining whether students’ knowledge was based on evidence or stereotypes. While most participants answered correctly items related to basic conceptual definitions (e.g., sex and gender perspective), correct response rates declined notably for more specific clinical content. Items 6, 7, 9, and 10 were the most difficult. For example, item 7, which addresses lipedema, a disease that affects an estimated 10% of women globally but remains under-recognized [20], was answered correctly by only 26.6%, with nearly 70% selecting “don’t know”. In contrast, in items 6, 9, and 10, many students chose incorrect answers rather than admitting a lack of knowledge, suggesting the presence of misconceptions.

Such misconceptions may be resistant to change and can negatively impact clinical decision-making [21]. For instance, Item 6 reflects the widespread “hormone myth”, the belief that most women become emotionally unstable during menstruation [22]. Regarding cardiovascular risk, Kuhn et al. [1] found that nurses were slower to triage women with acute coronary syndrome. In terms of pain perception (item 10), Wandner et al. [23] reported that health professionals consistently rated women’s pain as less intense and were more reluctant to administer opioids to them. Similarly, Prego-Jimenez et al. [24] found that gender stereotypes among healthcare professionals were associated with reduced credibility attributed to women’s pain reports.

This study has certain limitations. The data were collected from a specific population of nursing students, primarily white women from a single Spanish university, so caution is needed when generalizing the findings. Future studies should examine whether these misconceptions are also present among practicing health professionals and validate the scale in more diverse populations and healthcare contexts.

## 5. Conclusions

This study presents a practical and easy-to-use instrument for assessing gender-related health knowledge among nursing students. Although the scale demonstrates promising psychometric properties, further validation is required to confirm its applicability to medical students and practicing health professionals. At both educational and institutional levels, this tool may help identify knowledge gaps and misconceptions that should be addressed as modifiable risk factors for gender-biased healthcare.

The observed correlation between the Gender Knowledge Scale and the gender sensitivity subscale supports the notion that enhancing conceptual knowledge through education can promote more equitable and respectful clinical attitudes. In this context, instructional approaches should not only deliver new content but also support learners in unlearning misconceptions and reconstructing their understanding through a gender-sensitive lens. This process is essential to bridge the gap between policy-level commitments to gender equity in healthcare and their implementation in everyday clinical practice [4].

The systematic use of tools such as the Gender Knowledge Scale in health education and clinical environments represents a first step toward identifying areas of unawareness among both future and current professionals. Future research should examine the scale’s effectiveness in longitudinal studies tracking knowledge development over time, and in intervention studies evaluating whether addressing misconceptions leads to more gender-equitable clinical decisions. We believe the development and validation of this scale constitutes a valuable contribution to the pursuit of equal standards of care for all patients.

## Figures and Tables

**Table 1 nursrep-15-00187-t001:** Item-level descriptive statistics and standardized factor loadings from the confirmatory factor analysis.

Knowledge Item	Correct	Incorrect	Don’t Know	Loading
(1) Sex: conceptual definition	90.2	9.0	0.8	0.18
(2) Prescription of psychotropics	37.2	25.9	36.9	0.30
(3) Acute myocardial infarction	61.4	12.2	26.4	0.41
(4) Underestimated disease in one sex	65.1	9.5	25.4	0.53
(5) Gender perspective: Conceptual definition	74.1	3.9	22.0	0.55
(6) Premenstrual dysphoric disorder	13.4	40.9	45.7	0.28
(7) Lipedema	26.6	3.9	69.5	0.49
(8) Epidemiology: Data presentation	65.2	15.2	19.6	0.47
(9) Epidemiology: Mortality causes	20.5	58.0	21.5	0.12
(10) Pain	13.4	59.4	27.2	−0.04

**Table 2 nursrep-15-00187-t002:** Chi-square comparisons of correct and incorrect responses between high and low scorers on each knowledge item.

Knowledge Item	Low Scorers (*n* = 202)		High Scorers (*n* = 179)		*χ* ^2^	*V*
	Correct	Incorrect	Correct	Incorrect		
(1) Sex: conceptual definition	79.3	17.8	96.8	3.2	27.17 ***	0.28
(2) Prescription of psychotropics	17.2	34.3	64.7	12.8	82.73 ***	0.48
(3) Acute myocardial infarction	33.7	20.1	82.9	8.0	91.16 ***	0.51
(4) Underestimated disease in one sex	34.9	23.1	87.2	1.6	106.11 ***	0.55
(5) Gender perspective: Conceptual definition	48.5	8.3	92	0.5	82.47 ***	0.48
(6) Premenstrual dysphoric disorder	2.4	45.6	26.7	39.0	42.28 ***	0.35
(7) Lipedema	6.5	4.7	52.4	2.1	88.07 ***	0.50
(8) Epidemiology: Data presentation	32.0	27.2	88.8	5.9	121.98 ***	0.59
(9) Epidemiology: Mortality causes	6.5	62.7	38.5	42.2	50.90 ***	0.38
(10) Pain	6.5	60.4	21.4	58.3	19.31 ***	0.23

Note. Percentages do not sum to 100% because the “Don’t know” responses are not displayed. *** *p* < 0.001. *V* = Cramér’s V (effect size).

## Data Availability

The original dataset supporting the results of this study is openly available in Zenodo at https://doi.org/10.5281/zenodo.15176492 (accessed on 24 May 2025).

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
