# Peer review of "Health-Related Gender Knowledge: Scale Development and Validation in Spanish Nursing Students"

_nursrep, 2025, doi:10.3390/nursrep15060187_

Round 1

Reviewer 1 Report

Comments and Suggestions for Authors

Thank you for the opportunity to review this timely and relevant manuscript. The study presents the development and validation of a Gender Knowledge Scale to assess gender-related knowledge in Spanish nursing students. This work fills a gap by addressing the underassessment of gender knowledge in healthcare nursing students, and it offers a potentially valuable tool to complement existing measures of gender awareness. Here are our suggestions for improving the report. The merit of your work is not in question; however, some aspects need clarification or reformulation.

Title: Considering that the validation was developed exclusively with spanish nursing students, we suggest that the title clarifies the tool name and population, e.g., Development and Validation of the Gender Knowledge Scale in Spanish Nursing Students

Abstract: lacks a clear articulation of the research gap and does not present the results with sufficient precision or quantitative detail. The abstract should summarize the aim, methods, and results and convincingly justify the study by identifying the gap. The abstract must stand on its own. Additionally, the aim in the abstract is not the same as at the end of the introduction. The aim should always be the same in the manuscript.

Key-words: Consider adding “instrument development,” “psychometric validation,” and “nursing education” to enhance discoverability.

Introduction: The authors frame the study within the context of gender bias in healthcare and outline the relevance of gender knowledge in shaping attitudes and practices among healthcare professionals. The authors clearly articulate the gap in current measurement tools—specifically, the absence of a generalizable knowledge subscale in widely used instruments like N-GAMS. This gap justifies the development of a new scale. However, the theoretical framing could be deepened. While the manuscript references key constructs such as gender sensitivity and gender-role ideology, it would benefit from linking these to established cognitive or learning theories. For instance, integrating frameworks such as misconception theory, schema theory, or implicit bias theory could provide a richer conceptual basis for understanding how gender knowledge affects behavior and clinical judgment. This would also support the study’s claims about the educational implications of the instrument. Additionally, we suggest that the authors reflect on the denomination of the instrument as a complement. Can the tool be used on its own? We think that referring to the tool as a complement could diminish its impact and importance for the reader. We think it would be more appropriate to refer to this instrument as a tool that allows for a more comprehensive approach to issues related to gender in health professionals. The aim of the study, in the end of the introduction is "to create a Knowledge Scale for the Spanish nursing staff population 84 and analyse its psychometric properties". However, the study was developed with nursing students. We recognize that the authors' objective was to develop this instrument for health professionals, however, it has only been validated with nursing students, so the aim must be consistent with the methods. Suggestions for future research could include validating this instrument with health professionals, thus confirming its applicability to professionals. We don't think it's accurate to compare knowledge of gender issues between first-year nursing students and health professionals. In the case of health professionals, we can always theorize that questions that were very difficult for students may be significantly easier for professionals.

Methods: The methodology is generally robust and appropriate for the study objectives. The use of the Nominal Group Technique to generate scale content is a strength, ensuring that the items are informed by subject-matter experts and grounded in real-world relevance. The process is clearly described and includes steps to ensure content validity, such as iterative refinement and consensus-building among a multidisciplinary panel. However, the validity would benefit from factor analysis to assess the scales' dimensionality (despite acknowledging that only one construct was used). This is a standard psychometric procedure in scale development and is necessary to assess the dimensional structure of the scale. The current validation is limited to item difficulty analysis and correlations with related constructs, which, while informative, are not sufficient to establish construct validity. Please, include or justify the absence of factor analysis. Additionally, using the term "criterion validity" for item difficulty analysis may not be appropriate. (criterion validity is obtained by comparing its results to a known standard or "gold standard", Mokkink et al., 2010: https://pubmed.ncbi.nlm.nih.gov/20169472/). Please be clear of which references are being used to support the methodology. We suggest that the authors may use as a reference the work by Mokkink and colleagues (https://www.cosmin.nl/wp-content/uploads/COSMIN-methodology-for-content-validity-user-manual-v1.pdf). Concerning sample size (ln 92-94), the authors should provide the reference that supports this rationale. The scoring method (awarding one point for correct responses, subtracting 0.33 for incorrect responses, and assigning no value to “I don’t know”) is defensible, but its psychometric rationale should be better explained. The choice of −0.33 as a penalty appears arbitrary unless justified by a correction-for-guessing model. Including citations or precedent for this scoring strategy would strengthen this section. 

Results: are clearly presented, and the analysis is appropriate for the data collected. Item difficulty indices are calculated and show a meaningful distribution across easy, moderate, and difficult questions, which supports the scale’s ability to discriminate between different knowledge levels. The correlation between gender knowledge and gender sensitivity is statistically significant and of moderate size (rho = .30), supporting the hypothesized link between conceptual knowledge and attitudinal orientation. However, the lack of a significant relationship with gender-role ideology toward patients (rho = .01) deserves more reflection in the discussion. The analysis by academic year is insightful and reveals a pattern of improved scores between first- and second-year students, but no further progression in later years. This suggests a curriculum gap in sustained gender competence education. Differences by gender are non-significant, which is consistent with prior literature and well reported. One major omission is any form of internal consistency analysis. While the authors aim to measure a single construct—gender knowledge—it remains unclear whether the items cohere statistically as a scale. Including reliability coefficients would significantly strengthen the psychometric evaluation. 

Discussion: successfully interprets the main findings and situates them within existing research. The authors point out the lack of knowledge progression in later academic years, raising questions about the adequacy of gender-related instruction across the nursing curriculum. This is a crucial point, and the manuscript could benefit from concrete suggestions on how to better integrate gender competence training longitudinally. The discussion could be enriched by linking the findings more explicitly to broader educational or sociocognitive theories. Furthermore, the authors mention that the "I don’t know" option is a tool to prevent guessing, but they don't reflect on its possible pedagogical implications, e.g., as a metacognitive indicator of uncertainty or humility in clinical decision-making.

Conclusion: While the study is promising, the claim that this scale is “perfectly applicable to medical students and graduated health professionals” should be moderated unless supported by additional validation studies (please check what the suggestions were in the introduction section). The current sample is restricted to nursing students from a single geographic and cultural context. Future directions might include longitudinal studies to track changes in knowledge over time or intervention studies to assess how correcting misconceptions affects clinical practice.

Other suggestions: replace “Especifically” (ln 67) by “Specifically”; consider replacing "comprises a reliable instrument" (ln 203/204) by "is a reliable instrument". 

Author Response

Thank you very much for taking the time to review this manuscript. Please find the detailed responses below and the corresponding corrections highlighted in red in the re-submitted files.

Comments 1: Title: Considering that the validation was developed exclusively with Spanish nursing students, we suggest that the title clarifies the tool name and population, e.g., Development and Validation of the Gender Knowledge Scale in Spanish Nursing Students.

Response 1: Thank you for pointing this out, we have changed the tittle to Health-related gender knowledge: Scale development and validation in Spanish nursing students.

Comments 2: Abstract: lacks a clear articulation of the research gap and does not present the results with sufficient precision or quantitative detail. The abstract should summarize the aim, methods, and results and convincingly justify the study by identifying the gap. The abstract must stand on its own. Additionally, the aim in the abstract is not the same as at the end of the introduction. The aim should always be the same in the manuscript.

Response 2: Thank you very much. We have rewritten the abstract taking this into account.

Comments 3: Key-words: Consider adding “instrument development,” “psychometric validation,” and “nursing education” to enhance discoverability.

Response 3: Thank you very much for this suggestion. We have added those keywords to enhance discoverability.

Comments 4: The authors frame the study within the context of gender bias in healthcare and outline the relevance of gender knowledge in shaping attitudes and practices among healthcare professionals. The authors clearly articulate the gap in current measurement tools—specifically, the absence of a generalizable knowledge subscale in widely used instruments like N-GAMS. This gap justifies the development of a new scale. However, the theoretical framing could be deepened. While the manuscript references key constructs such as gender sensitivity and gender-role ideology, it would benefit from linking these to established cognitive or learning theories. For instance, integrating frameworks such as misconception theory, schema theory, or implicit bias theory could provide a richer conceptual basis for understanding how gender knowledge affects behavior and clinical judgment. This would also support the study’s claims about the educational implications of the instrument.

Response 4: We appreciate this thoughtful and constructive suggestion. We agree that incorporating a theoretical framework from cognitive and learning theories enriches the conceptual foundation of the study. In response, we have expanded the introduction to include schema theory, to contextualize how gender knowledge is cognitively structured and influences clinical decision-making. In addition, we refer to implicit bias theory to highlight how unconscious attitudes may shape healthcare professionals’ behaviors, particularly under conditions of uncertainty. We believe that this theoretical integration strengthens the study’s conceptual underpinnings and reinforces the educational relevance of the proposed instrument [lines 60-69 & 231-236].

Comments 5: Additionally, we suggest that the authors reflect on the denomination of the instrument as a complement. Can the tool be used on its own? We think that referring to the tool as a complement could diminish its impact and importance for the reader. We think it would be more appropriate to refer to this instrument as a tool that allows for a more comprehensive approach to issues related to gender in health professionals.

Response 5: Thank you for this suggestion. We agree that describing the Gender Knowledge Scale as a “complement” may inadvertently minimize its relevance. Accordingly, we have revised the manuscript to describe the scale as an instrument that allows for a more comprehensive assessment of gender-related issues in healthcare professionals. We believe this terminology more accurately reflects the tool’s value and aligns with the objective of enhancing gender awareness assessment by explicitly incorporating the dimension of conceptual knowledge, which is currently absent in widely used scales such as the N-GAMS [Changes have been made throughout the manuscript where appropriate.].

Comments 6: The aim of the study, in the end of the introduction is "to create a Knowledge Scale for the Spanish nursing staff population and analyse its psychometric properties". However, the study was developed with nursing students. We recognize that the authors' objective was to develop this instrument for health professionals, however, it has only been validated with nursing students, so the aim must be consistent with the methods. Suggestions for future research could include validating this instrument with health professionals, thus confirming its applicability to professionals. We don't think it's accurate to compare knowledge of gender issues between first-year nursing students and health professionals. In the case of health professionals, we can always theorize that questions that were very difficult for students may be significantly easier for professionals.

Response 6: We thank the reviewer for this important observation. We agree that the initial formulation of the study aim may have led to confusion regarding the target population. As suggested, we have revised the aim in the Introduction to accurately reflect the population used in this study, which consisted exclusively of nursing students. The revised sentence now reads: “The specific aim of this study was to develop the Gender Knowledge Scale and analyze its psychometric properties in a sample of Spanish nursing students” [88-89]. We also acknowledge the reviewer’s suggestion regarding future research, and we have incorporated this point into the Discussion section when addressing study limitations and directions for further validation among practicing health professionals [285-288].

Comments 7: Methods: The methodology is generally robust and appropriate for the study objectives. The use of the Nominal Group Technique to generate scale content is a strength, ensuring that the items are informed by subject-matter experts and grounded in real-world relevance. The process is clearly described and includes steps to ensure content validity, such as iterative refinement and consensus-building among a multidisciplinary panel.

Response 7: We thank the reviewer for their positive evaluation of the methodology. We appreciate the recognition of the use of the Nominal Group Technique and the efforts made to ensure content validity through a structured, expert-driven process

Comments 8: However, the validity would benefit from factor analysis to assess the scales' dimensionality (despite acknowledging that only one construct was used). This is a standard psychometric procedure in scale development and is necessary to assess the dimensional structure of the scale. The current validation is limited to item difficulty analysis and correlations with related constructs, which, while informative, are not sufficient to establish construct validity. Please, include or justify the absence of factor analysis.

Response 8: We thank the reviewer for this valuable suggestion. In response, we conducted a confirmatory factor analysis (CFA) using the WLSMV estimator and treating the ten dichotomous items as categorical variables. The one-factor model showed good fit indices (CFI = 0.928, TLI = 0.908, RMSEA = 0.025, SRMR = 0.075), supporting the unidimensional structure of the scale. These results have been included in the Data Analysis [168-170], Results [188-196], and Discussion [237-244] sections of the revised manuscript.

We acknowledge, however, that given the nature of the instrument—a multiple-choice knowledge test with heterogeneous item content and a correction-for-guessing scoring system—inter-item correlations are expected to be lower than in traditional attitude scales. For this reason, we complement the CFA with item-level analysis and evidence based on relationships with external constructs, in line with psychometric best practices for knowledge assessment tools.

Comments 9: Additionally, using the term "criterion validity" for item difficulty analysis may not be appropriate. (criterion validity is obtained by comparing its results to a known standard or "gold standard", Mokkink et al., 2010: https://pubmed.ncbi.nlm.nih.gov/20169472/). Please be clear of which references are being used to support the methodology. We suggest that the authors may use as a reference the work by Mokkink and colleagues (https://www.cosmin.nl/wp-content/uploads/COSMIN-methodology-for-content-validity-user-manual-v1.pdf).

Response 9: We appreciate the reviewer’s clarification regarding the correct use of psychometric terminology. We agree that the expression “criterion validity” was not appropriate in the context of our item-level analysis comparing responses from high- and low-scoring participants. Accordingly, we have revised the manuscript to refer to this analysis as “item discrimination analysis”, which more accurately reflects the procedure used. Additionally, we have revised our terminology throughout the Methods and Results sections to better align with established validation frameworks. We thank the reviewer for pointing us to the COSMIN guidelines and have considered this reference in framing our revisions [Changes have been made throughout the manuscript where appropriate].

Comments 10: Concerning sample size (ln 92-94), the authors should provide the reference that supports this rationale.

Response 10: We thank the reviewer for this observation. We have now added a reference to support the rationale behind our initial estimation of the minimum sample size, following classical recommendations on scale development (DeVellis, 2016) [97-99]. This reference has been included in the revised manuscript.

Comments 11: The scoring method (awarding one point for correct responses, subtracting 0.33 for incorrect responses, and assigning no value to “I don’t know”) is defensible, but its psychometric rationale should be better explained. The choice of −0.33 as a penalty appears arbitrary unless justified by a correction-for-guessing model. Including citations or precedent for this scoring strategy would strengthen this section. 

Response 11: We thank the reviewer for this comment. The scoring method we adopted is an adaptation of the correction-for-guessing approach commonly used in multiple-choice testing. While the strict formula would suggest a penalty of –0.5 (for 1 correct option out of 3 distractors), we opted for a less severe penalty (–0.33) to account for the fact that some incorrect answers may reflect partial understanding rather than random guessing. This decision aimed to balance discouraging uninformed guessing with recognizing students' tendency to rely on plausible—but incorrect—beliefs. We have now clarified this rationale in the Instruments section of the manuscript [111-115].

Comments 12: Results: are clearly presented, and the analysis is appropriate for the data collected. Item difficulty indices are calculated and show a meaningful distribution across easy, moderate, and difficult questions, which supports the scale’s ability to discriminate between different knowledge levels. The correlation between gender knowledge and gender sensitivity is statistically significant and of moderate size (rho = .30), supporting the hypothesized link between conceptual knowledge and attitudinal orientation. However, the lack of a significant relationship with gender-role ideology toward patients (rho = .01) deserves more reflection in the discussion. 

Response 12: We thank the reviewer for their positive assessment of the results and analyses. Regarding the non-significant correlation between gender knowledge and gender-role ideology toward patients (GRI-patient), we agree that this finding warrants further reflection. We have now added a paragraph in the discussion section to address this point and explore possible explanations [225-236].

Comments 13: The analysis by academic year is insightful and reveals a pattern of improved scores between first- and second-year students, but no further progression in later years. This suggests a curriculum gap in sustained gender competence education.

Response 13: We appreciate the reviewer’s interpretation regarding the lack of progression in knowledge scores beyond the second year. While we agree that this pattern may reflect a gap in sustained curricular emphasis, we note that the smaller number of participants in the third and fourth years limits the strength of this conclusion. For this reason, we chose to interpret these differences with caution in the manuscript [253-261].

Comments 14: Differences by gender are non-significant, which is consistent with prior literature and well reported. 

Response 14: We acknowledge the reviewer’s comment regarding the non-significant gender differences, which we reported in line with previous findings.

Comments 15: One major omission is any form of internal consistency analysis. While the authors aim to measure a single construct—gender knowledge—it remains unclear whether the items cohere statistically as a scale. Including reliability coefficients would significantly strengthen the psychometric evaluation. 

Response 15: We appreciate the reviewer’s observation regarding internal consistency. In response, we considered applying both Cronbach’s alpha and the Kuder-Richardson Formula 20 (KR-20), which are commonly used reliability indices. However, given the nature of our instrument—a multiple-choice knowledge scale with heterogeneous content, dichotomous scoring, and the inclusion of a “don’t know” option—inter-item correlations are expected to be low, and internal consistency indices may not accurately reflect the instrument’s quality or unidimensionality.

Instead, we focused on a confirmatory factor analysis (CFA), which supported the unidimensional structure of the scale with acceptable fit indices. Additionally, we provide item-level performance data and evidence based on relationships with external constructs, which together offer a robust psychometric foundation for the scale.

Comments 16: Discussion: successfully interprets the main findings and situates them within existing research. The authors point out the lack of knowledge progression in later academic years, raising questions about the adequacy of gender-related instruction across the nursing curriculum. This is a crucial point, and the manuscript could benefit from concrete suggestions on how to better integrate gender competence training longitudinally. The discussion could be enriched by linking the findings more explicitly to broader educational or sociocognitive theories.

Response 16: We thank the reviewer for this constructive and thoughtful feedback. We appreciate the positive assessment of the discussion. In response to the suggestions, we have now elaborated on the implications of the observed stagnation in knowledge by suggesting more explicit integration of gender competence training throughout the curriculum. Additionally, we have included a reference to broader educational theories, such as schema theory and implicit bias, to enrich the interpretation of our findings [225-236 & 245-260]

Comments 17: Furthermore, the authors mention that the "I don’t know" option is a tool to prevent guessing, but they don't reflect on its possible pedagogical implications, e.g., as a metacognitive indicator of uncertainty or humility in clinical decision-making.

Response 17: We agree that the “I don’t know” option may have pedagogical implications beyond its psychometric function, particularly as a metacognitive indicator of uncertainty or humility in clinical reasoning. Although we did not explore this dimension in the present study, we consider it a highly relevant line of inquiry for future research. It would be especially valuable to investigate whether responses to the “I don’t know” option correlate with psychological constructs such as prudence, self-awareness, epistemic humility, or conservatism, using complementary instruments alongside the Gender Knowledge Scale.

Comments 18: Conclusion: While the study is promising, the claim that this scale is “perfectly applicable to medical students and graduated health professionals” should be moderated unless supported by additional validation studies (please check what the suggestions were in the introduction section). The current sample is restricted to nursing students from a single geographic and cultural context. Future directions might include longitudinal studies to track changes in knowledge over time or intervention studies to assess how correcting misconceptions affects clinical practice.

Response 18: We thank the reviewer for this helpful comment. In the revised Conclusion, we have moderated the claims regarding the scale’s applicability and now state that additional validation is needed before the tool can be confidently used with medical students or practicing professionals. We also acknowledge that the current sample is limited to a specific academic and geographic context and highlight the need for future studies to confirm the scale’s generalizability [284-288]. Additionally, we have incorporated the reviewer’s suggestion by recommending that future research explore the use of the scale in longitudinal and interventional studies [305-309].

Other suggestions: replace “Especifically” (ln 67) by “Specifically”; consider replacing "comprises a reliable instrument" (ln 203/204) by "is a reliable instrument". 

We thank the reviewer for these language corrections. The suggested edits have been implemented in the revised manuscript. Additionally, the full manuscript has been professionally reviewed to ensure the overall quality and clarity of the English language [Changes have been made throughout the manuscript where appropriate].

We would like to sincerely thank the reviewers and the editor for their thorough and constructive feedback. We believe we have addressed all comments and suggestions, and we are confident that this revision has substantially improved the clarity, rigor, and overall quality of the manuscript.

Reviewer 2 Report

Comments and Suggestions for Authors
  1. Introduction

The introduction clearly states the importance of the topic, highlighting the problem of gender bias in health care and the need to address the lack of gender knowledge and sensitivity among health professionals.

The existing literature is reviewed, citing previous studies and World Health Organization documents that support the need to integrate a gender perspective in the training of health professionals.

The aim of the study is clearly defined: to develop a gender knowledge scale for Spanish nursing students and to analyze its psychometric properties.

  1. Materials and Methods

Participants: it would be interesting to better describe the sample, from a specific faculty in a specific location in Spain? On the other hand, being only nursing students, the survey may be biased towards that specific population. It seems necessary to extend the population to more health professionals, from more populations, from different professional fields (public employees, private employees, with more or less years of experience, etc.).

Instruments: to describe the psychometric properties of the Spanish adaptation of the Nijmegen Gender Awareness in Medicine Scale

Procedure: more description is needed on how the experts were selected, when and how the expert panel was conducted. How much time passed from the first to the second meeting, did the same professionals participate, where did they meet, etc. How were the participants selected to administer the survey?

How were participants selected to administer the survey, when, were they asked for consent, was the study explained to them? A lot of information is missing to understand how the whole study was conducted.

  1. Results

Line 167: “experimental results” and “experimental conclusions” is not adequate since you have not conducted an experimental design.

Analysis of the difficulty of the items: it would be interesting to analyze whether the errors made have been formulated by students who have just started their degree and have not yet received this knowledge or whether they have not been taught. There is an analysis by course in point 3.2. but I believe that it does not go into this aspect in depth.

Reliability, content validity and construct validity have yet to be presented.

  1. Conclusions: ok.

Author Response

Thank you very much for taking the time to review this manuscript. Please find the detailed responses below and the corresponding corrections highlighted in red in the re-submitted files

Comments 1: The introduction clearly states the importance of the topic, highlighting the problem of gender bias in health care and the need to address the lack of gender knowledge and sensitivity among health professionals.

The existing literature is reviewed, citing previous studies and World Health Organization documents that support the need to integrate a gender perspective in the training of health professionals.

The aim of the study is clearly defined: to develop a gender knowledge scale for Spanish nursing students and to analyze its psychometric properties.

Response 1: We thank the reviewer for their positive feedback and appreciation of our work. We are pleased that the importance of the topic, the relevance of the literature reviewed, and the clarity of the study’s aim were all recognized. These comments are encouraging and reinforce the value of our contribution to addressing gender bias in healthcare education.

Materials and Methods

Comments 2: Participants: it would be interesting to better describe the sample, from a specific faculty in a specific location in Spain?

Response 2: We have now specified in the manuscript that the participants were nursing students from the Faculty of Nursing, of the University of the Basque Country, Spain.

Comments 3: On the other hand, being only nursing students, the survey may be biased towards that specific population. It seems necessary to extend the population to more health professionals, from more populations, from different professional fields (public employees, private employees, with more or less years of experience, etc.).

Response 3: We thank the reviewer for this thoughtful observation. We agree that restricting the sample to nursing students may limit the generalizability of the findings. In response to a similar comment from the other reviewer, we have revised the objective of the study to clarify that the aim was to develop and validate the Gender Knowledge Scale specifically in a sample of Spanish nursing students. We have also acknowledged this limitation in the discussion and emphasized the need to extend the validation to a broader range of healthcare professionals in future studies.

Comments 4: Instruments: describe the psychometric properties of the Spanish adaptation of the Nijmegen Gender Awareness in Medicine Scale

Response 4: We thank the reviewer for pointing out this omission. In the revised manuscript, we have expanded the description of the S-NGAMS to include information about its psychometric properties. Specifically, we now note that the Spanish version of the scale has shown a two-factor structure consistent with the original version and demonstrated satisfactory internal consistency (α = 0.80 for Gender Sensitivity and α = 0.89 for GRI-patient) as reported in previous research. We also report that in the present study, internal consistency values were similar (α = 0.79 for GS and α = 0.88 for GRI-patient). This information has been added to the Instruments section of the manuscript.

Comments 5: Procedure: more description is needed on how the experts were selected, when and how the expert panel was conducted. How much time passed from the first to the second meeting, did the same professionals participate, where did they meet, etc.

Response 5: We thank the reviewer for this helpful suggestion. In response, we have expanded the Procedure section to include additional details about the expert panel process. The panel was formed through purposive sampling based on the participants’ backgrounds in health and gender training. All nine experts had experience in clinical practice and/or health education and had completed training in gender perspective. They were personally invited by the research team. The nominal group sessions were conducted online, with two sessions held two weeks apart. All nine participants attended both sessions. The sessions were moderated by a member of the research team experienced in qualitative research and expert group facilitation. These details have been added to the revised manuscript.

Comments 6: How were the participants selected to administer the survey? How were participants selected to administer the survey, when, were they asked for consent, was the study explained to them? A lot of information is missing to understand how the whole study was conducted.

Response 6: We thank the reviewer for highlighting the need for more clarity in the description of the participant recruitment process. We have now expanded the Procedure section to explain that participants were recruited through a convenience sampling approach. An invitation to participate in the study was sent via institutional email to nursing students from all academic years. The questionnaire was administered during the first semester of the academic year. Before starting the questionnaire, all participants received an information sheet explaining the purpose of the study, and informed consent was obtained electronically. The study was approved by the Ethics Committee of the University of the Basque Country (UPV/EHU). These clarifications have been incorporated into the revised manuscript.

Comments 7: Line 167: “experimental results” and “experimental conclusions” is not adequate since you have not conducted an experimental design.

Response 7: We thank the reviewer for pointing this out. The sentence in line 167 was part of the journal's template and was mistakenly left in the manuscript. It has now been removed.

Comments 8:  Analysis of the difficulty of the items: it would be interesting to analyze whether the errors made have been formulated by students who have just started their degree and have not yet received this knowledge or whether they have not been taught. There is an analysis by course in point 3.2. but I believe that it does not go into this aspect in depth.

Response 8: We appreciate this thoughtful observation. In the revised Discussion, we have addressed this point by reflecting on the possible curricular gaps that may explain the persistence of certain misconceptions or the lack of progression in knowledge beyond the second year. Specifically, we discuss how some of the most difficult items (e.g., those addressing pain perception or cardiovascular risk) may correspond to topics that are not yet introduced or are insufficiently addressed in the curriculum. We also emphasize the importance of reinforcing gender-related content longitudinally throughout the degree program to ensure sustained learning.

Comments 9: Reliability, content validity and construct validity have yet to be presented.

Response 9: We appreciate the reviewer’s observation. In the revised manuscript, we have provided evidence for all three aspects mentioned:

  • Content validity was addressed through the use of the Nominal Group Technique (NGT), involving a multidisciplinary panel of experts in gender and health to generate and refine the scale items.
  • Construct validity was supported by confirmatory factor analysis (CFA), which indicated acceptable fit for a unidimensional model of gender knowledge.
  • Reliability was evaluated through item discrimination analysis and further supported by external correlations with theoretically related constructs, such as Gender Sensitivity. As discussed in our response to point 7, we also considered internal consistency, but due to the heterogeneous nature of knowledge items, we opted to rely on factor structure and item-level analysis instead.

These elements together provide a comprehensive psychometric evaluation of the scale.

We would like to sincerely thank the reviewers and the editor for their thorough and constructive feedback. We believe we have addressed all comments and suggestions, and we are confident that this revision has substantially improved the clarity, rigor, and overall quality of the manuscript.

Round 2

Reviewer 1 Report

Comments and Suggestions for Authors

We congratulate the authors for their work.

Reviewer 2 Report

Comments and Suggestions for Authors

Authors have taken on board the comments and responded to the reviewer's queries, doing a good job. The effort to make these modifications is appreciated.